# Use of Renal Near-Infrared Spectroscopy and Urinary Neutrophil Gelatinase-Associated Lipocalin Monitoring as Indicators of Acute Kidney Injury in Pediatric Cardiac Surgery

**DOI:** 10.3390/jcm12062085

**Published:** 2023-03-07

**Authors:** Yoshihito Wakamatsu, Keisuke Nakanishi, Takanori Satoh, Shiori Kawasaki, Atsushi Amano

**Affiliations:** 1Department of Clinical Engineering, Juntendo University School of Medicine, 3-1-3, Hongo, Bunkyo-ku, Tokyo 113-8431, Japan; 2Cardiology Department, Pediatric and Congenital Cardiac Surgery Division, Juntendo University School of Medicine, 2-1-1, Hongo, Bunkyo-ku, Tokyo 113-8421, Japan

**Keywords:** acute kidney injury, cardiopulmonary bypass, neutrophil gelatinase-associated lipocalin, near-infrared spectroscopy, regional saturation of oxygen, pediatric cardiac surgery

## Abstract

Acute kidney injury (AKI) is a common complication following cardiac surgery under cardiopulmonary bypass (CPB) in children. A prospective study for examining urinary neutrophil gelatinase-associated lipocalin (NGAL) and renal near-infrared spectroscopy (NIRS) trends during AKI was conducted among pediatric patients undergoing cardiac surgery with CPB. Urinary NGAL showed a significant difference between intensive care unit admission (0 h) and 2 h post-admission (*p* < 0.001) and remained significant up to 4 h (*p* < 0.05). The renal NIRS in the AKI group showed a significant rate of decrease and lower values during the intraoperative period (*p* < 0.05). The cumulative median saturation of renal regional saturation of oxygen (rSO_2_) during CPB was 1637.5% min in the AKI group and 943.0% min in the non-AKI group. The median renal rSO_2_ scores at a reduction of 20% and 25% were significantly higher (*p* < 0.001) in the AKI group. Our results suggest that monitoring renal rSO_2_ scores and limiting their decline might be useful in preventing AKI. The combination of NGAL, renal rSO_2_, and renal rSO_2_ scores might be useful in the early diagnosis of AKI during pediatric cardiac surgery.

## 1. Introduction

Among the complications of cardiopulmonary bypass (CPB) surgery, the incidence of acute kidney injury (AKI) is as high as 47% in children. Of these, 67% of AKI cases occurred in children aged 2 years or below, with a younger age correlated with a higher incidence [1]. Furthermore, the onset of AKI is a known risk factor for prolonged intensive care unit (ICU) stay and increased mortality [2,3].

The standard definition of AKI is increased serum creatinine (sCr) levels or decreased urine output. A decrease in creatinine clearance signals its onset [3,4].

To facilitate early and reliable diagnosis, in 2011, the Kidney Disease: Improving Global Outcomes (KDIGO) guidelines defined the diagnostic criteria for AKI as decreased urine output for 6 h and increased sCr levels for 48 h. The usefulness of these criteria in a pediatric setting has been reported [5]. However, since most of these parameters reach their peak values 48 h after AKI onset, a reliable biomarker for the early detection of AKI is needed. Several studies have investigated potential early AKI biomarkers, such as neutrophil gelatinase-associated lipocalin (NGAL), Cystatin C, interleukin-18, renal injury molecule-1 (KIM-1), and others [6,7,8,9,10].

Due to the influence of maternal creatinine concentrations, the reference value of sCr in newborns changes daily and gradually increases as infants grow. Furthermore, its increase is slower at the onset of AKI compared to that in adults, and even a slight increase is associated with mortality [11]. Hence, diagnostic methods that can detect AKI earlier and are more sensitive than the existing methods are desirable in the pediatric setting.

We previously studied the correlation between AKI onset and urinary NGAL levels in pediatric patients undergoing CPB. We found that urinary NGAL levels were significantly elevated immediately after ICU admission in patients who developed postoperative AKI [12].

Other studies on AKI associated with pediatric cardiac surgery reported that the receiver operating characteristic (ROC) curves of urinary NGAL were 0.998 at 2 h after CPB and 1.000 at 4 h [13]. However, there are cases where NGAL values did not increase, despite postoperative AKI, or where NGAL values increased, despite the absence of postoperative AKI. These suggest that the values’ predictive performance is not consistent [14]. Similar results were observed in our study [12].

A previous study found that urinary NGAL was elevated at ICU admission in patients who developed AKI during the perioperative period. This suggests that many events that cause renal dysfunction might have occurred intraoperatively. Hence, it is necessary to evaluate intraoperative kidney perfusion for early AKI diagnosis. Therefore, we decided to use intraoperative near-infrared spectroscopy (NIRS) monitoring to evaluate blood flow to the kidney and measure renal oxygen saturation. NIRS is a non-invasive, continuous, and real-time method for measuring regional saturation of oxygen (rSO_2_) around the target organs. rSO_2_ captures the shift in the balance of oxygen supply and demand, reflecting tissue perfusion and metabolic status. NIRS can calculate the two-dimensional parameters of time elapsed below a set threshold and the value of the decline as the area under the curve (AUC). This allows the degree of desaturation to be quantified in terms of cumulative saturation (% min). The threshold for cumulative saturation can be set to the level of decline from the baseline or the level of decline at a set rate of decrease. In addition, since rSO_2_ provides real-time information about tissue perfusion—and the near-infrared light emitted by the sensors can reach several centimeters deep—we planned to examine its use in monitoring the perirenal region in children [15,16].

In this study, we examined the use of urinary NGAL and renal NIRS as indicators of AKI during the perioperative period of pediatric patients who underwent cardiac surgery under CPB.

## 2. Materials and Methods

### 2.1. Patient Population

This prospective observational study was approved (approval number 17-304) by the Center for Clinical Research and Clinical Trials of Juntendo University Hospital, which approves the clinical trial registry, and complies with the 1975 Declaration of Helsinki (revised 1983).

The study period ranged from June 2018 to January 2020, and all pediatric patients aged under 204 months with congenital heart disease undergoing cardiac surgery with CPB were eligible. Patients with written consent from their legal guardians were enrolled in the study. The exclusion criteria included cases requiring total circulatory arrest; beating-heart surgery; and before-surgery renal dysfunction, such as congenital renal disease.

Based on the 2016 Japanese Clinical Practice Guideline for AKI [17], which uses the KDIGO diagnostic criteria, AKI was classified according to sCr values, with those with stage 1 or higher considered in the AKI group [18].

### 2.2. Data Collection

Demographic and clinical data included patient background, duration of surgery, duration of CPB, duration of aortic cross-clamp, duration of postoperative ventilator use, PaO_2_/FiO_2_ (P/F) ratio before and after surgery, use of nitric oxide (NO), and serum lactate levels. sCr was evaluated before surgery and at 12 h and 36 h after surgery.

Disease and surgical factors were evaluated using the Risk Adjustment in Congenital Heart Surgery System score (RACHS-1 score), which evaluates surgical outcomes in cases of complex congenital heart disease [19].

### 2.3. CPB

The CPB priming volume was calculated according to institutional protocols, with the target flow rate based on a perfusion index of 2.6 L/min/m^2^. Body temperature was maintained at mild to moderate hypothermic levels. Blood was transfused to correct low hematocrit values (lower limit of 18%) whenever feasible. In all cases, modified ultrafiltration was performed immediately after CPB discontinuation.

CPB data collected from the automatic recording device ORSYS^®^ (Philips Co., Ltd., Amsterdam, The Netherlands) included dilution rate, perfusion index (PI = Perfusion flow/body surface area), hemoglobin level (HGB), hematocrit, and oxygen supply per body surface area (DO_2_i = 10 × pump flow (L/minute/m^2^) × HGB (mg/dL) × 1.34 × HGB saturation (%) + 0.003 × O_2_ tension (mmHg)).

### 2.4. Perioperative Management

Perioperative management was performed by an anesthesiologist, while postoperative management was handled by a cardiovascular surgeon. Hemodynamic management included the use of dopamine, dobutamine, noradrenaline, nitroglycerin, and other drugs. Ventilation management was mainly accomplished using pressure control ventilation with NO therapy as appropriate. Due to difficulties in monitoring urine output because of leakage caused by a catheter size mismatch and an irregular administration of diuretics, urine output was not included as an indicator.

### 2.5. Urinary NGAL

Urine collection was performed before surgery, at ICU admission (0 h), and at 1, 2, 4, 8, 12, 24, and 72 h. Urine samples were centrifuged at 500× *g* of relative centrifugal force for 5 min and frozen at −80 °C. Urinary NGAL measurements were performed using a chemiluminescence immunoassay with the immunoassay analyzer ARCHITECT i200SR^®^ (Abbott Japan Co., Ltd., Tokyo, Japan). Absolute values for urinary NGAL were used, without corrections for urinary creatinine or other parameters.

### 2.6. NIRS Monitoring

The INVOS 5100C (Medtronic, Minneapolis, MN, USA) was used for NIRS monitoring. Sensors were placed on the forehead and on the right perirenal area, with the renal sensor affixed to the ribs and midline of the pelvis (T10-L2), lateral to the spine, as reported by Ruf et al. [20]. NIRS data were collected every 2 s, from the time of entry into the operating room to the time of exit.

### 2.7. NIRS Monitoring Analysis

Analyses were performed using Medtronic’s INVOS software. The decrease in rSO_2_ from baseline (at the beginning of the surgery) was compared to its lowest value during surgery. Changes in rSO_2_ while on CPB were examined by comparing the mean, nadir, and rate of decrease from CPB initiation to the lowest rSO_2_ value. The decrease in rSO_2_ values during CPB was quantified and calculated to determine the cumulative saturation (defined as the renal rSO_2_ score). When the decrease in renal rSO_2_ scores from CPB initiation exceeded 20% and 25%, the amount of decline was integrated and calculated as the AUC (% min). The rSO_2_ score is {(baseline rSO_2_ − current rSO_2_ (%)) × time (min)}. The numerator of this score was changed from seconds to minutes to better accommodate longer time measurements.

### 2.8. Statistical Analysis

For statistical analysis, nonparametric tests, logistic regression analysis, and multiple regression analysis were performed using JMP^®^ 16.0.0 (SAS Institute Inc., Cary, NC, USA). The ROC curve was used to evaluate independent factors. The AUC was calculated, and the optimal cutoff value for discrimination of AKI onset was determined using the Youden index (defined as sensitivity + specificity − 1).

## 3. Results

### 3.1. Patient Groups

A total of 166 consecutive patients were eligible for inclusion in the study. Four patients were excluded following the exclusion criteria. Additionally, 48 patients were excluded due to missing values in each data set, resulting in a final number of 114 patients. The categorization of patients diagnosed with AKI is shown in Table 1. In total, 40 patients (35.1%) were diagnosed with AKI according to the KDIGO diagnostic criteria (observed until 72 h after ICU admission), with stage 1 AKI being the most common stage (60%), followed by stages 2 (30%) and 3 (10%). There was no postoperative death in the non-AKI and AKI groups. AKI determination was performed by a cardiovascular surgeon.

### 3.2. Patient Characteristics

Preoperative patient data are shown in Table 2, with the AKI group being younger in terms of age in months (*p* < 0.001) and having higher RACHS-1 scores (*p* < 0.001). The P/F ratio was lower in the AKI group (*p* < 0.05), and sCr levels were higher in the non-AKI group (*p* < 0.05).

### 3.3. Intraoperative and CPB Data

A comparison of intraoperative and CPB parameters is shown in Table 3. The duration of surgery (*p* < 0.05), duration of CPB, and the aortic cross-clamp time (*p* < 0.001) were significantly longer in the AKI group. The mean and minimum values of the perfusion index, HGB, and DO_2_i during CPB were not significantly different. However, the minimum value of DO_2_i was significantly lower in the AKI group (*p* < 0.05).

### 3.4. Postoperative Course

Table 4 shows the postoperative course of the patients. Intubation times were significantly longer in the AKI group (*p* < 0.001), and a significantly larger proportion of patients required NO therapy (*p* < 0.05). Additionally, sCr levels significantly increased in the AKI group 12 h after surgery (*p* < 0.001).

### 3.5. Urinary NGAL

NGAL measurement data were received after analysis had been completed by Abbott Japan Co., Ltd.

The time taken to receive them was approximately one to two months after submission.

Changes in urinary NGAL are shown in Table 5 and Figure 1. There was no significant difference between the two groups before surgery. However, a significant difference was observed from the time of ICU admission (0 h) to 2 h after (*p* < 0.001), and the difference persisted until 4 h later (*p* < 0.05).

### 3.6. Cerebral rSO_2_

Table 6 shows cerebral rSO_2_ levels. Cerebral rSO_2_ was significantly different between the two groups at the initiation of surgery, and the mean and lowest values during CPB were lower in the AKI group.

### 3.7. Renal rSO_2_

Renal rSO_2_ data are shown in Table 7. The rate of decrease of intraoperative rSO_2_ was significantly different between the AKI and non-AKI groups (*p* < 0.05). While there were significant differences in the mean (*p* < 0.001) and nadir values (*p* < 0.005) of rSO_2_ during CPB, there was no significant difference in the rate of decrease during CPB. In the ROC analysis, the lowest cutoff value of rSO_2_ during CPB was 57.0%, with a sensitivity of 62% and specificity of 31% (AUC 0.67).

Renal rSO_2_ scores during CPB are shown in Figure 2. The median renal rSO_2_ score during CPB was 943.0% min in the non-AKI group versus 1637.5% min in the AKI group. The median score at a 20% decrease was 8.0% min in the non-AKI group versus 85.5% min in the AKI group, and the median score at a 25% decrease was 1.0% min in the non-AKI group versus 32.5% min in the AKI group. There was a significant difference between the AKI and non-AKI group scores in the 20% and 25% decrease parameters (*p* < 0.001).

The median renal rSO_2_ score during CPB was 943.0% min in the non-AKI group versus 1637.5% min in the AKI group. The median score at a 20% decrease was 8.0% min for the non-AKI group and 85.5% min for the AKI group, and at a 25% decrease, the median score was 1.0% min for the non-AKI group and 32.5% min for the AKI group. A significant correlation was observed in both cases (*p* < 0.001).

## 4. Discussion

This study analyzed trends in urinary NGAL and renal rSO_2_ scores in patients younger than 204 months who developed postoperative AKI after cardiac surgery.

The RACHS-1 score showed that the AKI group included many patients with cyanotic heart disease who required more challenging surgery due to their younger age and complex cardiac malformations.

Previous studies reported that urinary NGAL is more sensitive than other biomarkers for detecting AKI [13,14,21]. Compared to our previous study involving 64 patients, perioperative urinary NGAL showed more sensitivity in predicting AKI in this study since the number of patients is higher [12]. Significant differences in urinary NGAL during ICU admissions suggest that factors that caused the elevated NGAL levels were present during the intraoperative period.

NIRS monitoring showed lower cerebral and renal rSO_2_ values in the AKI group than in the non-AKI group before the start of surgery. This is probably because the AKI group included more patients whose oxygen levels were already low, requiring respiratory support.

Under CPB conditions, patients are exposed to a non-physiological environment. In such environments, cerebral blood flow is significantly maintained by cerebrovascular autoregulation [22,23,24]. Although the kidneys are believed to exhibit regional autoregulation [25], the main cause of the decreased renal rSO_2_ was the decreased blood flow to the kidneys and other organs. This notion is based on the understanding that the kidneys are also considered to exhibit local autoregulatory functions [25], and the main cause of reduced kidney rSO_2_ is reduced blood flow to the organ due to blood shunting to the brain, which is sensitive to reduced blood flow.

NIRS monitoring is reported to produce artifacts and noise in the data, which can reduce its sensitivity and specificity [26,27,28]. There is also a lack of normative data and standardized definitions for renal ischemia, especially in the pediatric setting [29]. However, NIRS is non-invasive and provides continuous, real-time information about local tissue oxygen delivery and consumption in neonates [15,16]. Previous studies reported a significant correlation between changes in renal NIRS and the onset of AKI following CPB in critically ill children [20,22,30,31,32,33,34]. Furthermore, studies correlating renal blood flow with renal rSO_2_ also support this association [35].

NIRS is often used to monitor and detect changes, although it has no defined reference values. However, concerning patients with low values before surgery and the effects of artifacts, monitoring only absolute values or changes in values is inadequate. This is especially true in the CPB setting, where changes in hemodynamics, the partial pressure of oxygen, and HGB can be significant. The renal rSO_2_ score, which reflects cumulative changes in oxygen desaturation, should also be monitored to determine the baseline and the limit of acceptable desaturation levels.

Based on the results of this study, in addition to the conventional method of using only DO_2_i as an indicator of organ perfusion, we believe that monitoring the renal rSO_2_ score and limiting its decline might be useful in preventing AKI development. However, this study did not investigate the reasons for rSO_2_ decrease. In the future, we intend to further the research by examining factors, such as perfusion volume, blood pressure, partial pressure of oxygen, partial pressure of carbon dioxide, HGB, and body temperature, that might cause rSO_2_ decrease to determine methods that can prevent the onset of renal dysfunction.

This study had some limitations related to patient selection. Because this study included consecutive patients who had undergone surgery, we did not classify the patients by age in months as we did for neonates and infants. Moreover, we did not classify patients according to disease or disease severity, such as the presence or absence of cyanotic heart disease. In addition, cases in which it was impossible to collect urine were excluded. These included cases of leakage from urinary catheters, cases in which urinary drainage could only be seen in the urine bag tube, and cases in which the patient had to be switched to “nappies” early in the ICU. For these reasons, subgroup analysis and stratification were also deemed difficult due to the small number of valid cases. Additionally, hemolysis was not considered since no test was performed to assess it. Lastly, the renal rSO_2_ sensor is placed on the ‘perirenal’ area. It may not reliably reflect actual renal saturation, depending on body size, given the reach of near-infrared light. However, we also believe that it reflects abdominal blood flow, even if it does not directly reflect the renal. It may reflect the ischemic state of the major abdominal organs.

## 5. Conclusions

The combination of urinary NGAL, renal rSO_2_, and renal rSO_2_ scores might assist the early diagnosis of AKI after pediatric cardiac surgery. In addition to monitoring renal rSO_2_ scores, the routine use of cerebral and renal NIRS might reduce the incidence of AKI following cardiac surgery.

## Figures and Tables

**Figure 1 jcm-12-02085-f001:**
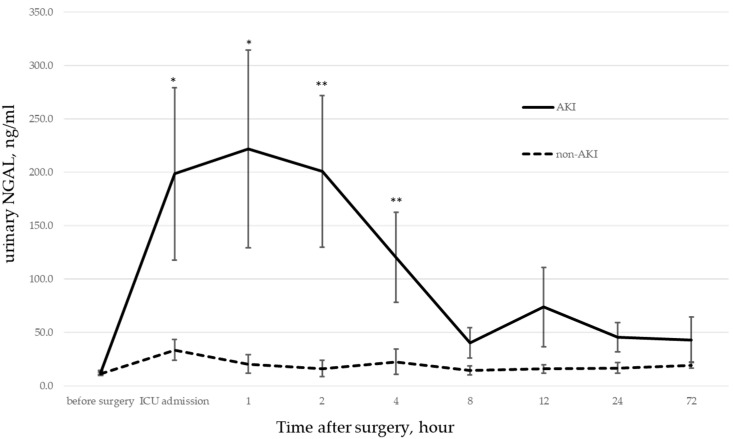
AKI vs. non-AKI NGAL. Changes in urinary NGAL in the AKI and non-AKI groups. There was no significant difference before surgery, but a strong significant difference was shown from the time of admission to the time of intensive care unit (ICU) admission (0 h) and 2 h later (*p* < 0.001 *), and the difference continued until 4 h later (*p* < 0.05 **).

**Figure 2 jcm-12-02085-f002:**
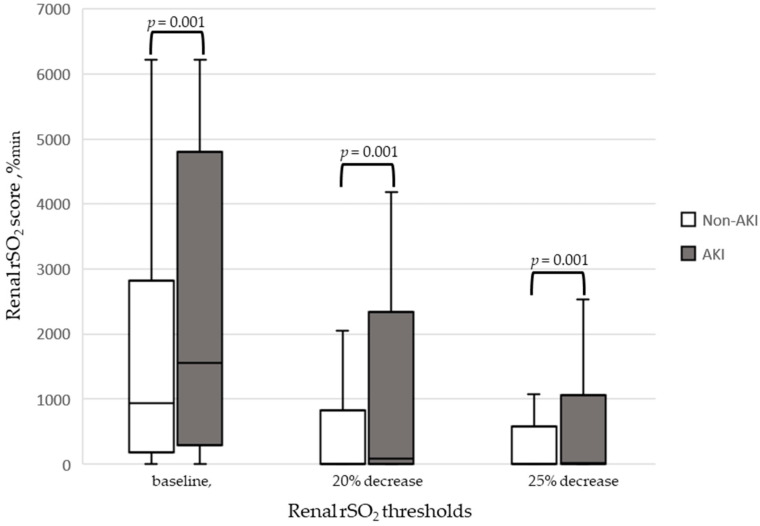
Comparison of cumulative saturation of rSO_2_ at baseline, 20% decrease, and 25% decrease.

**Table 1 jcm-12-02085-t001:** KDIGO AKI classification.

	*n* = 114 (AKI Groups *n* = 40; 35.1%)
AKI Stage1	*n* = 24; 60%
AKI Stage2	*n* = 12; 30%
AKI Stage3	*n* = 4; 10%

AKI: Acute kidney injury.

**Table 2 jcm-12-02085-t002:** Comparison of the preoperative clinical data.

	Non-AKI (*n* = 74)	AKI (*n* = 40)	*p*-Value
Age, month *	42.0 (14.0–92.3)	12.0 (6.8–22.5)	<0.001
Weight, kg *	13.6 (7.5–26.4)	7.5 (5.8–10.7)	0.041
Body surface area, m^2^ *	0.61 (7.5–26.4)	0.42 (0.33–0.52)	<0.001
Priming volume/Estimated blood volume	0.216 ± 0.069	0.235 ± 0.065	0.146
RACHS-1 score	2.0 ± 0.8	2.7 ± 0.9	<0.001
PaO_2_/FiO_2_ ratio	342.1 ± 166.7	270.3 ± 177.5	0.032
Serum lactate, mmol/L	1.01 ± 0.68	1.04 ± 0.80	0.844
Serum creatinine, mg/dL	0.36 ± 0.14	0.28 ± 0.09	<0.001
HGB, g/dL	11.2 ± 1.4	11.5 ± 1.7	0.408
Platelets, ×10^4^/μL	20.1 ± 6.5	25.0 ± 6.2	0.986

Data are presented as median (interquartile range) *. Data are presented as mean ± standard deviation. RACHS-1: Risk Adjustment in Congenital Heart Surgery System.

**Table 3 jcm-12-02085-t003:** Comparison of intraoperative clinical data.

	Non-AKI (*n* = 74)	AKI (*n* = 40)	*p*-Value
OP time, min	204.1 ± 76.8	243.2 ± 72.1	0.014
CPB time, min	77.6 ± 35.6	109.1 ± 38.3	<0.001
Aortic cross-clamp time, min	38.8 ± 28.4	63.2 ± 36.9	<0.001
Mean PI, L/min/m^2^	2.55 ± 0.30	2.53 ± 0.30	0.306
Minimum PI, L/min/m^2^	2.29 ± 0.27	2.23 ± 0.30	0.166
Minimum temperature (rectal), °C	33.7 ± 2.8	32.6 ± 1.9	0.026
Mean HGB, g/dL	8.3 ± 1.2	9.0 ± 1.4	0.231
Minimum HGB, g/dL	7.6 ± 1.2	7.4 ± 1.0	0.231
Mean DO_2_i, mL/min/m^2^	284.0 ± 64.6	281.4 ± 64.5	0.505
Minimum DO_2_i, mL/min/m^2^	249.8 ± 53.9	225.7 ± 42.2	0.008

Data are presented as mean ± standard deviation. OP: operation, CPB: cardiopulmonary bypass, PI: perfusion index, HGB: hemoglobin, HCT: hematocrit, DO_2_i: oxygen delivery index.

**Table 4 jcm-12-02085-t004:** Comparison of postoperative clinical data.

	Non-AKI (*n* = 74)	AKI (*n* = 40)	*p*-Value
Duration of mechanical ventilation, min	16.6 ± 34.6	40.2 ± 35.3	<0.001
P/F ratio, mmHg	462.8 ± 230.8	415.1 ± 179.5	0.27
Inhaled NO therapy	8 cases (10.8%)	11 cases (27.5%)	0.026
Serum lactate, mmol/L	1.30 ± 0.63	1.51 ± 0.72	0.114
Post OP 12 h serum creatinine, mg/dL	0.39 ± 0.14	0.50 ± 0.21	<0.001
Post OP 36 h serum creatinine, mg/dL	0.33 ± 0.13	0.33 ± 0.20	0.7771

Data presented as mean ± standard deviation. P/F: PaO_2_/FiO_2_, NO: nitric oxide, OP: operation.

**Table 5 jcm-12-02085-t005:** Performance of urinary NGAL in AKI diagnosis.

Time after Surgery (h)	AUC	Cut-Off	Sensitivity	1-Specificity
before surgery	0.55	12.5	0.45	0.22
ICU admission	0.73	13.1	0.72	0.48
1	0.79	8.7	0.74	0.55
2	0.79	8.1	0.66	0.51
4	0.74	5.4	0.79	0.48
8	0.68	9.5	0.57	0.30
12	0.68	6.6	0.79	0.33
24	0.73	13.8	0.62	0.37
72	0.53	6.2	0.76	0.17

AUC: area under the curve, ICU: intensive care unit.

**Table 6 jcm-12-02085-t006:** Cerebral rSO_2_ clinical outcomes.

	Non-AKI (*n* = 74)	AKI (*n* = 40)	*p*-Value
Before surgery cerebral rSO_2_ (%)	69.5 (60.0–75.0)	61.0(55.8–70.0)	0.031
Intraoperative cerebral rSO_2_ decrease (%)	27.6 (22.4–35.1)	31.5 (21.4–39.2)	0.06
CPB mean cerebral rSO_2_ (%)	73.0 (64.0–77.0)	61.0 (50.0–74.0)	0.013
CPB mean cerebral rSO_2_ decrease (%)	14.9 (8.1–28.4)	20.8 (12.9–31.7)	0.06
CPB cerebral rSO_2_ nadir (%)	54.0 (47.0–62.0)	48.5 (43.8–57.3)	0.039

Median (interquartile range). rSO_2_: regional oxygen saturation, CPB: cardiopulmonary bypass.

**Table 7 jcm-12-02085-t007:** Cumulative saturation of CPB Renal rSO_2_ score decrease.

	Non-AKI (*n* = 74)	AKI (*n* = 40)	*p*-Value
Before surgery renal rSO_2_ (%)	73.5 (46.0–85.0)	57.5 (34.8–79.0)	0.094
Intraoperative renal rSO_2_ decrease (%)	13.4 (6.8–29.0)	27.5 (17.9–40.1)	0.012
CPB mean renal rSO_2_ (%)	77.0 (56.0–87.0)	60.0 (34.0–79.0)	0.001
CPB renal rSO_2_ nadir (%)	71.0 (42.5–81.0)	53.0 (26.0–70.0)	0.005
CPB renal rSO_2_ decrease (%)	4.4 (0.0–11.7)	7.7 (0.8–17.9)	0.253

Median (interquartile range). rSO_2_: regional oxygen saturation, CPB: cardiopulmonary bypass.

## Data Availability

Data sharing not applicable. No new data were created or analyzed in this study. Data sharing is not applicable to this article.

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
