# Peer review of "Use of Renal Near-Infrared Spectroscopy and Urinary Neutrophil Gelatinase-Associated Lipocalin Monitoring as Indicators of Acute Kidney Injury in Pediatric Cardiac Surgery"

_jcm, 2023, doi:10.3390/jcm12062085_

Round 1

Reviewer 1 Report

The authors conducted a prospect study, in which they investigated the trends in urinary NGAL and renal rSO2 in children with postoperative AKI and non-AKI after cardiac surgery. In the study, they demonstrated that the combination of urinary NGAL and renal rSO2 might be helpful for the early diagnosis of postoperative AKI. The topic is interest and clinically important. The following are my comments to improve the manuscript.

Major comments

1. Page2 Line 84

Could you explain why patients aged under 204 months were included in this study? As the authors mentioned in the introduction, the renal pathophysiology of neonate is different from that of teenagers. Age and RACHS-1 scores could be a confounder. Thus, I would recommend for the authors to conduct subgroup analysis.

2. Is this prospect study registered in a registry? This is because to avoid making change in outcomes the authors should make it sure that the outcomes are determined before analysis.

3. Page4 Line 149-152

To investigate the predicting ability of NGAL for AKI, NGAL should be elevated before the diagnosis of AKI. I recommend for the authors to clarify that when these subjects were diagnosed as having AKI and who diagnosed AKI.

Minor comments

1. Page3 Line 125-127

Is there any reason why the sensors were placed on the right perirenal area?

2. Page 5 Line 175

Could you explain what is “intubation times”?

Author Response

We would like to thank Reviewer 1 for the time and efforts invested in reviewing our manuscript and for providing comments, which have considerably helped us improve our manuscript. We have made revisions based on your comments and have provided our point-by-point responses below. We hope our responses and revisions appropriately address your comments.

Major comments

Page2 Line 84

Could you explain why patients aged under 204 months were included in this study? As the authors mentioned in the introduction, the renal pathophysiology of neonate is different from that of teenagers. Age and RACHS-1 scores could be a confounder. Thus, I would recommend for the authors to conduct subgroup analysis.

→Thank you for your constructive comment. In Japan, adulthood is reached at 18 years; patients younger than this age were included in the study. We will conduct a subgroup analysis in a subsequent paper.

Is this prospect study registered in a registry? This is because to avoid making change in outcomes the authors should make it sure that the outcomes are determined before analysis.

→Yes. Our study is registered in the registry in Juntendo University. We have inserted the following sentence in the manuscript,

“this prospective study is registered in Juntendo University Registry  (approval number 17-304).”

Page4 Line 149-152

To investigate the predicting ability of NGAL for AKI, NGAL should be elevated before the diagnosis of AKI. I recommend for the authors to clarify that when these subjects were diagnosed as having AKI and who diagnosed AKI.

→Thank you once again. We have added this information to the text (Page5, 186-189).

Minor comments

Page3 Line 125-127

Is there any reason why the sensors were placed on the right perirenal area?

→Yes. It is because of the relation between the position of the INVOS body and the position of the body of the electrocautery unit.

Page 5 Line 175

Could you explain what is “intubation times”?「Time spent on the ventilator.」

→Intubation time is the duration of intubation.

Reviewer 2 Report

Abstract:

Cumulative median saturation of renal regional saturation of oxygen (rSO2) was higher in the AKI group than in the non-AKI group. Please give more intuitive explanation. 

“The median renal rSO2 scores at a reduction of 20% and 25% were significantly lower in the AKI group.” This sentence is opposite to that of Results page 7, lines 221-223: “The median score at 20% decrease was 8.0%min for the non-AKI group and 85.5%min for the AKI group, and at 25% decrease, the median score was 1.0%min for the non-AKI group and 32.5%min for the AKI group”, as well as to Figure 2. Please explain. 

Stay in use of the abbreviation for NGAL in the last sentence.

Introduction.

Page 2, line 58-59. Please insert a citation where similar results were observed.

Methods.

Page 3, 2.3. CPB, Please give more explanation on perfusion index and DO2i in detail, i.e., how they were obtained or calculated from your CPB data. DO2i is the abbreviation for “oxygen supply for body surface area”? Please be specific.

Page 3, 2.6 NIRS Monitoring, 

The authors put the renal sensor at the ‘perirenal area’ to monitor renal regional saturation during surgery.

However, can the sensors attached to the ‘perirenal area’ reliably reflect the renal saturation?

What is the range o f depth of the target tissue from the sensor that the regional saturation of the tissue can be reliably monitored?

How they define the ‘perirenal area’?

Please provide references for that.

Page 3, 2.7. NIRS Monitoring Analysis, 

Lines 134-137, Please explain more in detail about calculation of cumulative saturation of rSO2, which is depicted in Figure 2. 

Did you add values of every min or every 2-sec? Indicate any reference on the “rSO2 scores” if it exists. 

Results

Table 2, Please indicate preoperative GFR, if data is available. 

Table 3, Please define PI (perfusion index) and DO2i (oxygen delivery index) and explain how they were obtained or calculated in Methods section. 

Figure 1. Please indicate the units of X and Y axis. Also, indicate ‘ICU admission’ at 0 and thereafter, and indicate ‘pre’ more specifically. I would suggest replacing ‘pre’ with ‘before surgery’. 

Figure 1. Why don’t you indicate significantly different values (0-4 hours postop) such as with asterisks in the graph?

Figure 1 legends, correct “dif ference” to “difference”

Page 7, lines 220-221: The median renal rSO2 scores are higher in the non-AKI group compared to the AKI group. Does that mean cumulative saturation of renal rSO2 is higher in the non-AKI group than in the AKI group? Please explain this. 

Same as for the median score at 20% and 25% decrease (Lines 221-224).

Discussion.

The authors should acknowledge that renal rSO2 sensor attached to the “perirenal area”, which was not defined exactly where it was, may not have reliably reflected the real renal saturation in patients with various ages and body sizes. I suggest that the authors should discuss (or provide rebuttal) on this issue and add limitation regarding this issue. 

Author Response

We would like to thank Reviewer 2 for the time and efforts invested in reviewing our manuscript and for providing comments, which have considerably helped us improve our manuscript. We have made revisions based on your comments and have provided our point-by-point responses below. We hope our responses and revisions appropriately address your comments.

Abstract:

Cumulative median saturation of renal regional saturation of oxygen (rSO2) was higher in the AKI group than in the non-AKI group. Please give more intuitive explanation. 

→We have changed the sentence to “Renal regional saturation of oxygen (rSO2) was higher in the AKI group than in the non-AKI group.”

“The median renal rSO2 scores at a reduction of 20% and 25% were significantly lower in the AKI group.” This sentence is opposite to that of Results page 7, lines 221-223: “The median score at 20% decrease was 8.0%min for the non-AKI group and 85.5%min for the AKI group, and at 25% decrease, the median score was 1.0%min for the non-AKI group and 32.5%min for the AKI group”, as well as to Figure 2. Please explain. 

→Thank you once again. This has been corrected in the text.

Stay in use of the abbreviation for NGAL in the last sentence.

→Thank you for pointing this out. We have corrected this.

Page 2, line 58-59. Please insert a citation where similar results were observed.

→We have added a reference citation.

Methods.

Page 3, 2.3. CPB, Please give more explanation on perfusion index and DO2i in detail, i.e., how they were obtained or calculated from your CPB data. DO2i is the abbreviation for “oxygen supply for body surface area”? Please be specific.

→We have added an explanation on Page3, 104-108.

Page 3, 2.6 NIRS Monitoring, 

The authors put the renal sensor at the ‘perirenal area’ to monitor renal regional saturation during surgery.

However, can the sensors attached to the ‘perirenal area’ reliably reflect the renal saturation?

What is the range o f depth of the target tissue from the sensor that the regional saturation of the tissue can be reliably monitored?

How they define the ‘perirenal area’?

Please provide references for that.

→ Although this is an assumption, the depth is about 3 cm. Furthermore, we believe that it is easier for the sensor to measure kidney rSO2 in children because of the close proximity of the kidneys to the body surface. We have cited a supporting paper in page4, line 127.

Page 3, 2.7. NIRS Monitoring Analysis, 

Lines 134-137, Please explain more in detail about calculation of cumulative saturation of rSO2, which is depicted in

→We have added this on Page3, line 138-140.

Figure 2. 

Did you add values of every min or every 2-sec? Indicate any reference on the “rSO2 scores” if it exists. 

→Yes. Reference [20]

This was an automatic analysis with special software. The numerator of this score was changed from seconds to minutes to better accommodate longer measurements.

Results

table 2, Please indicate preoperative GFR, if data is available. 

The AKI diagnosis is made by comparison with the preoperative Cre value and that it is only a reference value as the patient is a child.

Patients with preoperative abnormal renal function are excluded.

Table 3, Please define PI (perfusion index) and DO2i (oxygen delivery index) and explain how they were obtained or calculated in Methods section. 

→We have added this on Page3, line 105-108.

Figure 1. Please indicate the units of X and Y axis. Also, indicate ‘ICU admission’ at 0 and thereafter, and indicate ‘pre’ more specifically. I would suggest replacing ‘pre’ with ‘before surgery’. 

→Thank you for indicating this. This has been corrected in the text.

Figure 1. Why don’t you indicate significantly different values (0-4 hours postop) such as with asterisks in the graph?

→Thank you. We have corrected this.

Figure 1 legends, correct “dif ference” to “difference”

→This has been corrected. 

Page 7, lines 220-221: The median renal rSO2 scores are higher in the non-AKI group compared to the AKI group. Does that mean cumulative saturation of renal rSO2 is higher in the non-AKI group than in the AKI group? Please explain this. 

→If we only consider rSO2 values before surgery and during CPB, the non-AKI group had higher values.

There was a significant difference in the rate of decline during surgery. The quartiles were 17.9-40.1. We do not know what percentage drop is dangerous.

Therefore, we examined the cumulative saturation, which indicates the amount of decline. The results show that the AKI group had a higher amount of decline than the non-AKI group.

This is also the case with the median score at 20% and 25% decrease (Lines 219-224).

→This has been corrected. 

Discussion.

The authors should acknowledge that renal rSO2 sensor attached to the “perirenal area”, which was not defined exactly where it was, may not have reliably reflected the real renal saturation in patients with various ages and body sizes. I suggest that the authors should discuss (or provide rebuttal) on this issue and add limitation regarding this issue. 

→ Added this point to the Restrictions section (page 9, lines 291-292).

Round 2

Reviewer 1 Report

Thank you for revising the manuscript. The authors answered all of my comments point by point. However, did not seem to be revised the manuscript satisfactorily. The following are my comments to the authors.

Thank you for your constructive comment. In Japan, adulthood is reached at 18 years; patients younger than this age were included in the study. We will conduct a subgroup analysis in a subsequent paper.

The authors did not revise the manuscript. If subgroup analysis or stratification is difficult to conduct, the authors need to explain why.

Yes. Our study is registered in the registry in Juntendo University. We have inserted the following sentence in the manuscript, this prospective study is registered in Juntendo University Registry (approval number 17-304).”

“Juntendo University Registry” an ethics committee not a clinical trial registry.

Thank you once again. We have added this information to the text (Page5, 186-189).

I would want to know when the subjects were diagnosed having AKI. This is because, for instance, AKI after 1 months of surgery due to sepsis is not associated with post-operative NGAL and renal rSO2 during surgery.

Intubation time is the duration of intubation.

The duration of mechanical ventilation might be better for expression.

Author Response

Thank you for your constructive comment. In Japan, adulthood is reached at 18 years; patients younger than this age were included in the study. We will conduct a subgroup analysis in a subsequent paper.

→The authors did not revise the manuscript. If subgroup analysis or stratification is difficult to conduct, the authors need to explain why.

below comment has been added to the limitation. 'For these reasons, subgroup analysis and stratification were also deemed difficult due to the small number of valid cases.'(Page 9, line 508-510)

Yes. Our study is registered in the registry in Juntendo University. We have inserted the following sentence in the manuscript, “this prospective study is registered in Juntendo University Registry (approval number 17-304).”

→ “Juntendo University Registry” an ethics committee not a clinical trial registry.

 This prospective observational study was approved by the Juntendo University Hospital Clinical Research and Trial center at Juntendo University School of Medicine (17-304) and complies with the Helsinki Declaration of 1975 (revised 1983) '(Page 2, line 84-86).

Thank you once again. We have added this information to the text (Page5, 186-189).

→I would want to know when the subjects were diagnosed having AKI. This is because, for instance, AKI after 1 months of surgery due to sepsis is not associated with post-operative NGAL and renal rSO2 during surgery.

Additional, observed until 72 hours after ICU admission. '(Page ,4 line 256-257)

Intubation time is the duration of intubation.

→The duration of mechanical ventilation might be better for expression.

→Changed as specified. '(Page 5, Table4)

Reviewer 2 Report

I appreciate authors revising their manuscript fairly well.

I would recommend to reinforce the limitations of this study, as I mentioned previously, that perirenal attachment of the NIRS sensor does not necessarily reflect kidney saturation especially in this heterogenous pediatric population with varying ages and body sizes. The authors just add one sentence that it may not reflect actual kidney saturation as the last sentence of the limitations (page 9, lines 290-291). Please add more details carefully.

Author Response

I would recommend to reinforce the limitations of this study, as I mentioned previously, that perirenal attachment of the NIRS sensor does not necessarily reflect kidney saturation especially in this heterogenous pediatric population with varying ages and body sizes. The authors just add one sentence that it may not reflect actual kidney saturation as the last sentence of the limitations (page 9, lines 290-291). Please add more details carefully.

→Additional 「Lastly, the renal rSO2 sensor is located 'perirenal'. It may not reliably reflect actual renal saturation depending on body size, given the reach of . near infrared light However, we believe that it reflects abdominal blood flow, even if it does not directly reflect the renals. It may reflect the ischemic state of the major abdominal organs.」'(Page 9, line 511-515)